# Changes in the Arbuscular Mycorrhizal Fungal Community in the Roots of *Eucalyptus grandis* Plantations at Different Ages in Southern Jiangxi, China

**DOI:** 10.3390/jof10060404

**Published:** 2024-06-04

**Authors:** Yao Jiang, Xiao-Yong Mo, Li-Ting Liu, Guo-Zhen Lai, Guo-Wei Qiu

**Affiliations:** 1College of Forestry and Landscape Architecture, South China Agricultural University, Guangzhou 510642, China; jiangyy@scbg.ac.cn (Y.J.); motree@163.com (X.-Y.M.); 2Jiangxi Academy of Forestry, Nanchang 330000, China; laiguozhen@jxlky.cn; 3Jinpenshan Forest Farm, Xinfeng 341600, China; xfjpslc@163.com

**Keywords:** stand ages, soil properties, *Paraglomus*, *Glomus*

## Abstract

Eucalyptus roots form symbiotic relationships with arbuscular mycorrhizal (AM) fungi in soil to enhance adaptation in challenging environments. However, the evolution of the AM fungal community along a chronosequence of eucalypt plantations and its relationship with soil properties remain unclear. In this study, we evaluated the tree growth, soil properties, and root AM fungal colonization of *Eucalyptus grandis* W. Hill ex Maiden plantations at different ages, identified the AM fungal community composition by high-throughput sequencing, and developed a structural equation model among trees, soil, and AM fungi. Key findings include the following: (1) The total phosphorus (P) and total potassium (K) in the soil underwent an initial reduction followed by a rise with different stand ages. (2) The rate of AM colonization decreased first and then increased. (3) The composition of the AM fungal community changed significantly with different stand ages, but there was no significant change in diversity. (4) *Paraglomus* and *Glomus* were the dominant genera, accounting for 70.1% and 21.8% of the relative abundance, respectively. (5) The dominant genera were mainly influenced by soil P, the N content, and bulk density, but the main factors were different with stand ages. The results can provide a reference for fertilizer management and microbial formulation manufacture for eucalyptus plantations.

## 1. Introduction

Arbuscular mycorrhizal (AM) fungi, forming symbiotic relationships with the roots of approximately 80% of vascular plants globally, act as conduits for material transport between plants and soil [1]. While deriving organic nutrients from plants and soil, AM fungi enhance the absorption of water and mineral elements by plants in the soil [2,3], aiding in stress alleviation [4,5]. The significant application potential of AM fungi in enhancing forest quality, mitigating adverse afforestation effects, and contributing to ecological restoration has garnered increasing attention [6,7,8].

The establishment of mycorrhiza in eucalyptus has been demonstrated in many studies [9,10,11], and this symbiotic relationship plays an important role in promoting plant growth and boosting productivity [12]. The influence of AM fungi on plant growth is contingent on their community composition [13,14]. Finding functional symbiotic AM strains and further manufacturing microbial formulations are key steps toward commercial application. However, the reliability of the traditional morphological identification method is low [15,16]. At present, the understanding of the AM fungal community composition of eucalyptus species roots in the natural environment is incomplete. High-throughput sequencing technology based on DNA extraction can effectively improve the accuracy of fungal identification [17], providing a way to supplement and improve the AM fungal classification system of eucalyptus roots.

Some studies have found that the community structure of AM fungi in artificial forests tends to change regularly as stand ages increase [18,19]. This is partly due to the changes in the physiological characteristics and growth states of plants at different stand ages, which affect the ability of AM fungi to colonize roots [20] and the allocation of carbon from plants to AM fungi [21], thereby altering the AM fungal community structure. Additionally, as plantations mature, the availability of soil resources for AM fungi also changes continuously. Research has shown that factors such as the soil pH value [22] and nutrient content [23,24] significantly influence the composition of AM fungal communities. Despite these findings, the relationships among the AM fungal community, tree growth, and soil properties remain unclear.

The hilly region of southern Jiangxi, China constitutes a vital segment of the red soil belt in southern China, where severe soil erosion prevails due to overexploitation and recurrent heavy rainfall [25,26]. *Eucalyptus grandis* W. Hill ex Maiden is a prominent afforestation tree with robust adaptability, rapid growth rates, and high productivity, which can create huge economic and ecological benefits for the local area [27,28]. Understanding the changes in the AM fungal community composition of the *E. grandis* root and its influencing factors can provide a reference for the management of plantations in the hilly areas of southern Jiangxi. Consequently, this study focused on six monoculture *E. grandis* plantations of different ages (2, 3, 5, 6, 7, and 9 years old) in the hilly region of southern Jiangxi Province. We investigated tree growth, analyzed soil physicochemical properties, determined root AM fungal colonization, and identified the composition of the AM fungal community by high-throughput sequencing. Subsequently, we analyzed the correlations among tree growth status, soil properties, and AM fungal communities in *E. grandis* plantations of different ages, and we established a structural equation model to reveal the causality of the soil–tree–mycorrhiza system. Our objective was to address the following specific questions: Is there a difference in AM fungal community structure in *E. grandis* roots of different ages? If so, are changes in the AM fungal community structure with age related to changes in soil properties?

## 2. Materials and Methods

### 2.1. Experimental Site and Sampling

This study was conducted at plantations in southern Jiangxi Province, China (114°32′–115°59′ E, 24°42′–26°33′ N; elevation between 220 and 360 m). This study area lies on the southern edge of Central Asia and is characterized by a subtropical humid monsoon climate, with an annual average temperature of 18.9 °C and a frost-free period of 287 d. The mean annual precipitation is 1605 mm, mostly between April and June (China Meteorological Data Service Center). The soil at the study site has a loamy texture and is colonized by native grasses and forbs (such as *Miscanthus floridulus* (Labill.) Warburg ex K. Schumann, *Dicranopteris dichotoma* (Thunb.) Bernh., and *Rubus innominatus* S. Moore). *E. grandis* plantations are widely planted in the study area, and the stand spacing is usually 1.67 m × 4 m.

All planted areas with relatively consistent slopes (25–35°) and uniform management modes were selected. Besides adding base fertilizer in the year of planting, these plantations received a topdressing for three consecutive years. The ratio of N, P, and K in these fertilizers is 15:15:15. These woodlands are effectively protected by cutting prohibitions and fire prevention. The chronosequence of *E. grandis* plantations includes six ages (2, 3, 5, 6, 7, and 9 years old). Two to three stands were selected for each stand age, and 20 m × 20 m plots were set for each stand, constituting 17 plots in total. The average tree height and diameter at breast height (DBH) were measured in each plot.

Three sampling points with a distance of more than 10 m between them were randomly selected for each sample plot, and soil samples with a depth of more than 0–20 cm were drilled with a ring knife for bulk density measurement. At the same time, 500 g soil samples were taken from each sampling point for other physical and chemical properties’ measurements. All samples were stored in a foam box with an ice pack and transported to the laboratory. Five trees were randomly selected from each plot, the fine roots were carefully collected along the taproots of the trees, and then the fine roots collected in the plot were mixed into one sample. The fine roots were cleaned with running water in the laboratory, passed through a 0.2 mm sieve, and dried with a towel. The soil properties of *Eucalyptus grandis* plantations with different stand ages are shown in Table 1.

### 2.2. Soil Properties

The soil water content was measured using the AOAC [29] method, and the bulk density was calculated. Soil pH was measured using the glass electrode method. Total nitrogen (N), total phosphorus (P), and total potassium (K) contents were quantified using Kelvin-distillation titration, sodium hydroxide fusion–Mo-Sb colorimetric, and sodium hydroxide fusion–flame atomic absorption spectrometry, respectively [30]. The available N, P, and K contents were quantified by the alkali diffusion method, hydrochloric acid–ammonium fluoride extraction–Mo-Sb colorimetric, and ammonium acetate extraction–flame atomic absorption spectrometry. In addition to the common soil elements mentioned above, boron (B) is widely believed to be involved in the growth of eucalyptus trees, especially in root nutrition [31,32]. The available B content was measured using the curcumin colorimetry method [33].

### 2.3. AM Fungi Colonization

According to the method of Phillips and Hayman [34], partial root samples were alkalified in 10% KOH solution for 3 h (water bath at 90 °C), then bleached with 3% H_2_O_2_ for 20 min, acidified with 2% HCl for 30 min, and finally dyed with 0.1% acid fuchsia at 60 °C for 1 h. The stained root samples were made into slides, and AM fungi colonization was measured [35]. One hundred views for each slide were randomly observed using a compound microscope at 200× magnification. The presence of hyphae or arbuscules was recorded with each view, and the results were calculated using the following formulas: hyphal colonization rate = the number of views with a hyphal structure/total number of views; arbuscular colonization rate = the number of views with arbuscular structure/total number of views.

### 2.4. Sequencing and Data Analysis

#### 2.4.1. Sequencing

DNA was extracted from eucalyptus roots using an EZNA Gel Extraction Kit (Omega, Irving, TX, USA). DNA integrity and purity were monitored on 1% agarose gels. DNA concentration and purity were measured using the NanoDrop One at the same time. 18SrRNA genes of distinct regions (V4/V5) were amplified using a specific primer (528F and 706R) with a 12 bp barcode [36,37]. PCR reactions, containing 25 μL 2x Premix Taq, 1 μL each primer (10 mM), and 3 μL DNA (20 ng/μL) template in a volume of 50 µL, were amplified by thermocycling: 5 min at 94 °C for initialization; 30 cycles of 30 s denaturation at 94 °C, 30 s annealing at 52 °C, and 30 s extension at 72 °C; followed by 10 min final elongation at 72 °C. Three replicates per sample and each PCR product of the same sample were mixed, with the PCR instrument BioRad S1000. The length and concentration of the PCR product were detected by 1% agarose gel electrophoresis. Samples with bright main strips could be used for further experiments. PCR products were mixed in equidense ratios according to the GeneTools Analysis Software (Version 4.03.05.0, SynGene). Then, the mixture of PCR products was purified with an EZNA Gel Extraction Kit. Sequencing libraries were generated using a NEBNext^®^ Ultra™ DNA Library Prep Kit for Illumina^®^ (New England Biolabs, Ipswich, MA, USA) and index codes were added. The library quality was assessed on a Qubit@ 2.0 Fluorometer and Agilent Bioanalyzer 2100 system. At last, the library was sequenced on an IlluminaHiseq2500 platform, and 250 bp paired-end reads were generated.

#### 2.4.2. Sequencing Data Processing

Quality filtering on the paired-end raw reads was performed under specific filtering conditions to obtain high-quality clean reads according to the Trimmomatic (V0.33, http://www.usadellab.org/cms/?page=trimmomatic, accessed on 23 April 2024) quality-controlled process. Paired-end clean reads were merged using FLASH (V1.2.11, https://ccb.jhu.edu/software/FLASH/, accessed on 23 April 2024) according to the relationship of the overlap between the paired-end reads; when at least 10 of the reads overlapped the read generated from the opposite end of the same DNA fragment, the maximum allowable error ratio of the overlap region was 0.1, and the spliced sequences were called Raw Tags. Sequences were assigned to each sample based on their unique barcode and primer using Mothur software (V1.35.1, http://www.mothur.org, accessed on 23 April 2024), after which the barcodes and primers were removed and the effective Clean Tags were obtained.

#### 2.4.3. OTU Cluster and Species Annotation

Sequence analyses were performed by Usearch software (V8.0.1517, http://www.drive5.com/usearch/, accessed on 23 April 2024). Sequences with ≥97% similarity were assigned to the same operational taxonomic units (OTUs). The most frequently occurring sequence was extracted as representative for each OTU and screened for further annotation. During clustering, a Usearch can remove the chimera sequence and singleton OTU at the same time. For each representative sequence, the Silva database (https://www.arb-silva.de/, accessed on 23 April 2024) was used to annotate taxonomic information (setting the confidence threshold to default to ≥0.5). We removed the polluted OTUs and obtained the number of effective tags and an OTU taxonomy synthesis information table for the final analysis. OTU abundance information was normalized using a standard sequence number corresponding to the sample with the fewest sequences. Subsequent analyses were performed based on these outputted normalized data. Alpha diversity was applied in analyzing the complexity of species diversity for a sample through 5 indices: observed species ($I_{\mathrm{obs}}$, the number of OTUs measured in the sample), Chao1 ($I_{\mathrm{chao}1}$, see Equation (1)), dominance ($I_{\mathrm{dom}}$, see Equation (2)), Shannon index ($I_{\mathrm{shn}}$, see Equation (3)), and Simpson index ($I_{\mathrm{sim}}$, see Equation (4)). All these indices in our samples were calculated with QIIME (V1.9.1) and displayed with R software [38].
(1)$$I_{\mathrm{chao}1}=I_{\mathrm{obs}}+\frac{n_{1}(n_{1}-1)}{2(n_{2}+1)}$$
where $n_{1}$ is the number of OTUs containing only one sequence; ${\mathrm{and}\;n}_{2}$ is the number of OTUs containing only two sequences.
(2)$$I_{\mathrm{dom}}=\sum p_{i}^{2}$$
where $p_{i}$ represents the proportion of the *i*-th OTU in the total OTUs.
(3)$$I_{\mathrm{shn}}=-\sum_{i=1}^{I_{\mathrm{obs}}}\frac{n_{i}}{N}\ln\frac{n_{i}}{N}$$
(4)$$I_{\mathrm{sim}}=1-\frac{\sum_{i=1}^{I_{\mathrm{obs}}}n_{i}(n_{i}-1)}{N(N-1)}$$
where $n_{i}$ is the number of OTUs containing *i*-th sequences. N is the number of all sequences.

### 2.5. Statistical Analysis

One-way ANOVA and multiple comparisons were used to detect the differences in tree height and DBH of trees at different ages. The changes in soil properties and mycorrhizal colonization rates with age were analyzed using regression analysis. To test the significance of the variation in AM fungal community composition with different stand ages, an analysis of similarities was conducted using the function “ANOSIM” in the R package VEGAN [39]. The relative abundance data for AM fungi were visualized using non-metric multidimensional scaling (implemented using the R package VEGAN) to elucidate the dissimilarities in AM fungal community composition across sites along the chronosequence [39]. Pearson’s correlation analysis evaluated the relationship between age-affected soil properties and AM fungal attributes. Structural equation modeling was used to show a causal relationship between these variables, and the significance of the standardized regression weights was determined [40,41]. Structural equation modeling was performed using the LAVAAN package in R [42]. In addition, we used the Chi-square/df, *p*-value, and CFI index to examine the overall model fit. The model fit the data well when the Chi-square/df was less than 2, the *p*-value was more than 0.05, or the CFI index was more than 0.9.

## 3. Results

### 3.1. Growth Statuses of Eucalypts with Different Stand Ages

The one-way ANOVA showed that there were significant differences in tree height (*p* < 0.01) and DBH (*p* < 0.01) among different age groups. The results of multiple comparisons using the LSD method (Figure 1) showed that the mean tree height and DBH first increased rapidly and then slowly with an increase in stand age. There were significant differences in tree height between 2 and 3 years old and between 3 and 5 years old and in DBH between 3 and 5 years old.

### 3.2. Soil Properties of Eucalypt Plantations with Different Stand Ages

The regression analysis showed that with the increase in stand age, the contents of total P (R^2^ = 0.37, *p* < 0.05) and total K (R^2^ = 0.42, *p* < 0.05) in the soil decreased first and then increased. The contents of these two elements were the lowest at 5 and 6 years old, respectively. Soil pH, bulk density, and the contents of available B, total N, N, P, and K did not change significantly (Figure 2).

### 3.3. AM Fungal Colonization Rates in the Roots of Eucalypts with Different Stand Ages

The regression analysis showed that the hyphal (R^2^ = 0.40, *p* < 0.05) and arbuscular (R^2^ = 0.45, *p* < 0.05) colonization rates in roots of eucalypts decreased first and then increased with the increase in stand age and that the arbuscular colonization rate decreased slightly at 9-year-old stands (Figure 3). The highest hyphal and arbuscular colonization rates were found in 2-year-old stands, reaching 0.81 and 0.58, respectively. The lowest hyphal and arbuscular colonization rates were 0.62 and 0.35, respectively, at 5 years old.

### 3.4. AM Fungal Identification and Community Composition with Different Stand Ages

Through high-throughput sequencing, more than 50,000 eligible 18S sequences were obtained. Of these sequences, 2103 OTUs were annotated after being classified according to the 97% similarity criterion. Among these, 1750 OTUs were identified at the genus level. Through the identification of the AM fungal community in roots, a total of one phylum, one class, four orders, seven families, and nine genera were found (Figure 4). The AM fungal communities were composed predominantly of members of the genus *Paraglomus* (70.1%), followed by *Glomus* (21.8%). Along the chronosequence, the relative abundance of *Paraglomus* in AM fungal communities showed a significant difference (F = 4.76, *p* < 0.05), and that of *Glomus* showed a highly significant difference (F = 8.58, *p* < 0.01). The 6-year-old stands were significantly different from other ages, with the lowest *Paraglomus* abundance (43%) and the highest *Glomus* abundance (52%). ANOSIM showed that the AM fungal community composition was significantly affected by stand age (R^2^ = 0.52, *p* < 0.01). Non-metric multidimensional scaling analysis showed clear clustering of the AM fungal communities in the roots of *E. grandis* at different stand ages (Figure 5). However, the diversity of the AM fungal community was not affected by stand age in this study.

### 3.5. Relationships among Plant Growth, Soil Properties, and AM Fungal Attributes

The correlational relationships between the age-affected soil properties and AM fungal attributes are shown in Figure 6. The relative abundances of *Glomus* and *Paraglomus* were significantly negatively correlated. The relative abundance of *Paraglomus* was significantly negatively correlated with soil total P and available N content and significantly positively correlated with soil bulk density. The relative abundance of *Glomus* was positively correlated with soil total P and the available N content but negatively correlated with total K and the bulk density. Both the hyphal and arbuscular colonization rates were positively correlated with the soil total K. In addition, the arbuscular colonization rate was positively correlated with soil pH and negatively correlated with total N and the available N content. Both the Shannon index and Chao1 index were positively correlated with soil available B.

A structural equation model was used to analyze the relationship between the dominant genera, tree growth, and soil properties. The model analysis results indicated the following (Figure 7): (1) In 2–6 years of eucalypt plantations, the soil total P, total N, and available N contents were continuously consumed with the increase in tree height. The decrease in total P and available N increased the abundance of *Paraglomus* but decreased the abundance of *Glomus*. (2) The soil total P content at 7–9 years of eucalypt plantations gradually accumulated with the increase in DBH, resulting in decreased *Paraglomus* abundance and *Glomus* abundance. The growth of DBH also directly reduced *Paraglomus* abundance, while the decrease in soil bulk density increased *Paraglomus* abundance.

## 4. Discussion

Our analysis of the growth statuses of eucalypt plantations of different ages showed that the growth of eucalypts underwent rapid development and a slow-down stage. In the rapid development stage (2–6 years old), the main change in soil properties was that the total P and total K contents gradually decreased with the increase in stand age, and the lowest values appeared in the 5- or 6-year-old stands. This may have been due to the high metabolism of the trees during this period and the high consumption of nutrients [43]. At the same time, the contents of available N and P in the soil gradually increased. With the increase in stand age, the increase in soil microorganisms and root exudates in the understory promoted the conversion of nutrients from insoluble to soluble [44]. The reason for the decrease in the available K content is that during rapid development, the maintenance of leaf photosynthetic capacity [45] and the improvement of water use efficiency [46] make the consumption of K much higher than that of other elements. Regarding AM fungal properties, hyphal and arbuscular colonization rates decreased gradually with the increase in stand age and dropped to the lowest for 5-year-old stands. This was because the survival of AM fungi depends on the carbon source provided by plants [47]. In the period of vigorous growth, the nutrient allocation of trees is more inclined to go to the aboveground parts, while the carbon allocation of the underground parts is gradually reduced [48]. During the growth-slowing stage (7–9 years old), the total P and K contents in the soil gradually increased with the stand age. Soil microorganisms and root exudates decomposed the increasing understory litter and returned to the soil as nutrients [49]. At the same time, hyphal and arbuscular colonization rates increased gradually because the increased numbers of lateral and fibrous roots provided more sites for AM fungi to colonize [50]. Currently, most eucalypt plantations in the study area have a five-year rotation period, meaning that the stand is cut down before nutrients are returned to the soil as litter. Incomplete nutrient cycling reduces the net nutrient supply and the amount of mycelium required for the next growth in the soil, increasing the dependence on fertilizer [51,52]. Therefore, proper crop rotation expansion is an important measure for the sustainable management of eucalypt plantations.

Through high-throughput sequencing of AM fungi in roots of *E. grandis*, one phylum, one class, four orders, seven families, and nine genera were identified. Among these, the major genera, such as *Acaulospora*, *Archaeospora*, *Gigaspora*, *Glomus*, *Paraglomus*, and *Scutellospora*, have been shown to form symbiotic structures with eucalypts [53,54]. *Ambispora*, *Claroideoglomus*, and *Diversispora* are also found in eucalypt plantations [55,56]. Most have studies confirmed the dominant position of *Glomus* in the AM fungal community in eucalyptus roots [57,58]. However, in this study, the relative abundance of *Paraglomus* (70.1%) was higher than that of *Glomus* (21.8%) in the AM fungal community in the roots of *E. grandis* in southern Jiangxi, China. One reason could be that the spores of *Paraglomus* are very similar to those of some *Glomus*, meaning the presence of *Paraglomus* species may go unrecognized by traditional spore-based AM fungal identification techniques [59]. Other studies have suggested that Paraglomerales is usually distributed in patches, which may explain the large differences in fungal abundance detected at different locations [60]. The relative abundance of *Paraglomus* was significantly superior in almost all stand ages of *E. grandis* plantations, which may also be related to the host tree’s preference for AM fungi [61]. The relative abundance of *Glomus* was greater than that of *Paraglomus* only at 6 years old, possibly because *Glomus* is more competitive than other genera when stand growth slows and the carbon allocation to underground parts decreases [62]. This kind of genus is characterized by rapid colonization and various colonization routes [63]. Other studies have also shown that plants can adapt to new environments by increasing symbiotic structures with more competitive dominant species [64,65]. Our study revealed significant differences in the compositions of the AM fungal community in the roots of eucalypts with different stand ages, which was consistent with the findings of other scholars [11,66,67]. In this study, the diversity of AM fungal communities did not change significantly over time, which is related to maintaining the diversity of aboveground plant communities [68]. In addition to AM fungi, *E. grandis* roots also form symbiotic structures with ectomycorrhizal fungi [69]. Further studies should consider the contribution of different mycorrhizal types and their community composition to ecosystem functioning.

The correlation analysis between soil properties and AM fungal attributes under the influence of age indicated that the mycorrhizal colonization rate was affected by soil pH, total K, total N, and the available N content. In other studies, the rhizosphere environment has also been found to affect the mycorrhizal colonization process [70], but the specific mechanism is still unclear. Both the Shannon index and Chao1 index were positively correlated with available soil B, indicating that the diversity of AM fungi may be mainly affected by soil trace elements. Correlation analysis revealed the basic relationship between soil properties and AM fungal attribute variables. On this basis, the structural equation model further demonstrated the relationship between tree growth, soil properties, and dominant genera at different stages. The total P content in the soil was always the main factor limiting the relative abundance of *Paraglomus*. The relative abundance of *Glomus* was mainly affected by the available soil N content at the rapid growth stage (2a–6a) and was significantly negatively correlated with soil bulk density at the growth slow-down stage (7a–9a). This is consistent with the view of most studies that AM fungi generally have a high affinity for P and N elements [71,72]. Numerous studies have shown that P is the main limiting factor in plantations in southern China [73,74]. AM fungi can absorb P from the soil through the mycelium network, which helps trees grow in a poor P environment [75]. Studies have shown that the mycelia of *Glomus* tend to be very thin and that these heavily extended and dispersed epiphytic mycelia can stabilize soil aggregates under a low soil bulk density [76] and improve soil water relations. Our structural equation model revealed the effects of soil properties on the AM fungal community at different growth stages. The findings are particularly important for optimizing fertilization strategies and introducing AM fungal inoculants at specific growth stages. In conclusion, changes in the AM fungal community composition in tree roots can be attributed to changes in soil properties caused by tree growth. Ideally, future research should focus on building a specific understanding of the functional properties of AM fungal species, to enable the precise use of AM fungal agents.

## 5. Conclusions

This study delved into the growth dynamics and soil properties of monospecific *E. grandis* at varying ages, employing high-throughput sequencing technology to unravel the composition of the root AM fungal community. Through comprehensive statistical analyses, we scrutinized the interplay between growth statuses, soil properties, and the successional patterns of the root AM fungal community in eucalypt plantations of diverse ages. The findings revealed that with the increase in stand age, the soil total phosphorus (P) and total potassium (K) decreased first and then increased; the rate of AM colonization decreased first, reached the lowest in 5a stands, and then increased; there was no significant change in the community diversity of root AM fungi, but a significant change in community composition, especially in the relative abundance of two dominant genera, *Glomus* and *Paraglomus*. It could be seen from the structural equation model that the soil total P content limited the abundance of *Paraglomus*. The abundance of *Glomus* was mainly affected by the soil-available nitrogen (N) content in the rapid-growing period (2–6 years old) and the soil bulk density in the slow-growing period (7–9 years old). In conclusion, prolonged cultivation of monospecific *E. grandis* significantly influenced the accumulation of soil K and N, thereby inducing shifts in the structure of the AM fungal community, which was primarily evident in the relative abundance alterations of *Glomus* and *Paraglomus*. The results of this study are of great significance for optimizing fertilization measures at different growth stages and improving tree productivity in nutrient-poor environments of eucalyptus plantations. Subsequent research endeavors should prioritize exploring the functional disparities among different AM fungal species to unlock the full potential of AM fungi in driving vegetation restoration and enhancing forest quality.

## Figures and Tables

**Figure 1 jof-10-00404-f001:**
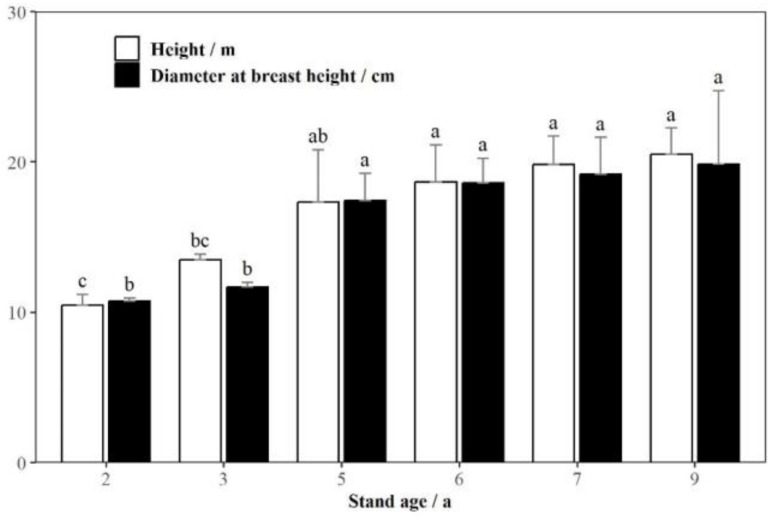
Heights and diameters at breast height of *Eucalyptus grandis* with different stand ages. Note: Different letters show significant differences at the *p* < 0.05 level according to the LSD test.

**Figure 2 jof-10-00404-f002:**
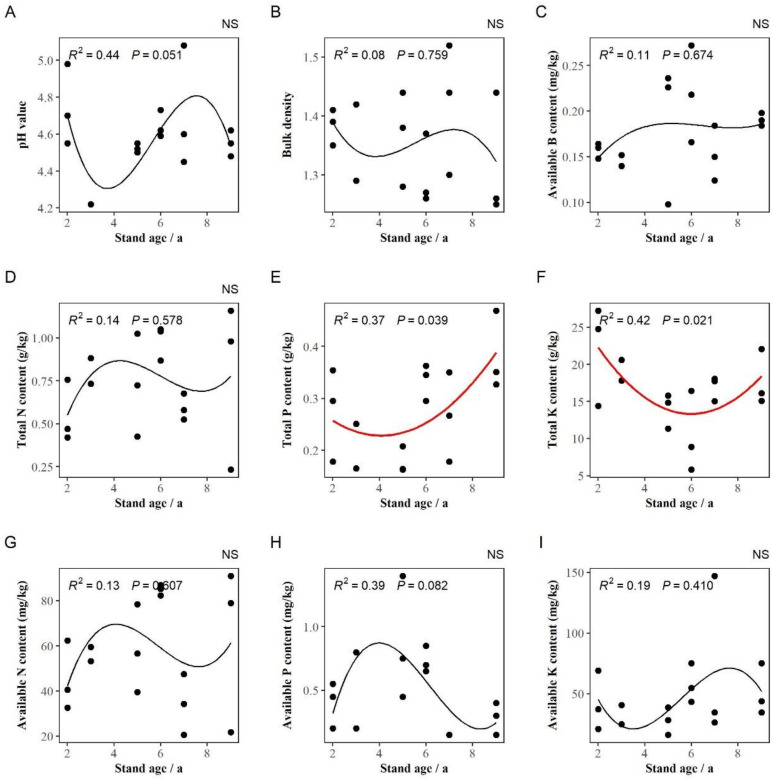
Changes in soil properties in *Eucalyptus grandis* plantations with different stand ages. (**A**–**I**) represent the change in soil pH value, bulk density, available boron content, total nitrogen content, total phosphorus content, total potassium content, available nitrogen content, available phosphorus content, and available potassium content, respectively. Note: The R^2^ and *p*-values at the top of the images represent the correlations and significance levels of the regression analysis, respectively. NS represents non-significant regressions with different stand ages (*p* ≥ 0.05). The red curves represent significant regression results.

**Figure 3 jof-10-00404-f003:**
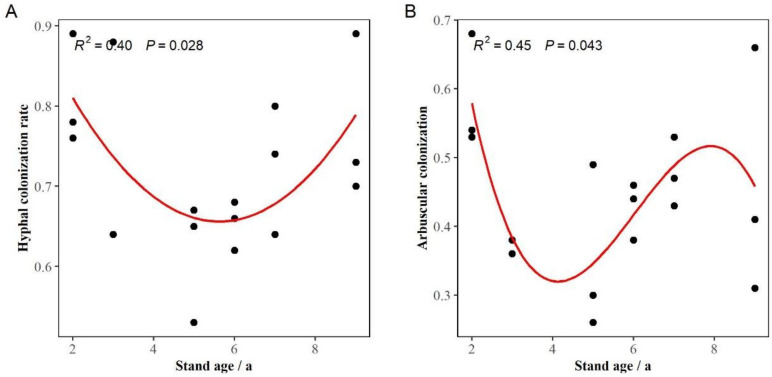
Changes in AM fungal colonization rate in the roots of *Eucalyptus grandis* with different stand ages. (**A**,**B**) represent the change in hyphal and arbuscular colonization rates, respectively. Note: The R^2^ and *p*-values at the top of the images represent the correlations and significance levels of the regression analysis, respectively. The red curves represent significant regression results.

**Figure 4 jof-10-00404-f004:**
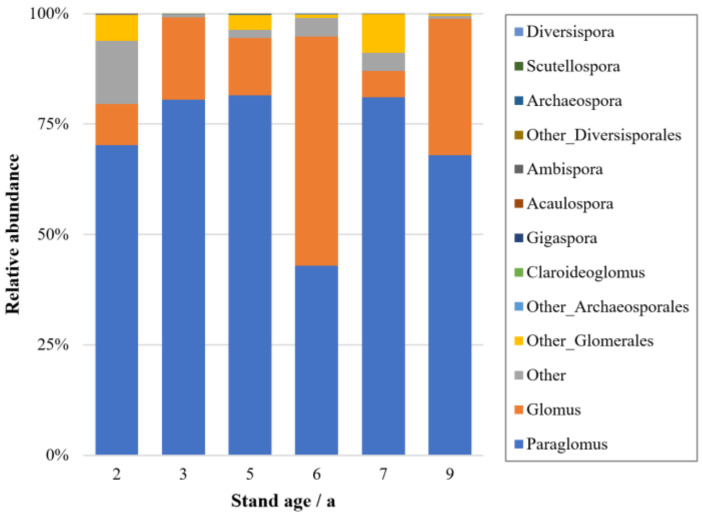
Proportions of AM fungal genera in roots of *Eucalyptus grandis* with different stand ages based on the relative abundance of genera. Note: Some genera with very low relative abundance are labeled only with their orders, and “Other_” is added as the prefix.

**Figure 5 jof-10-00404-f005:**
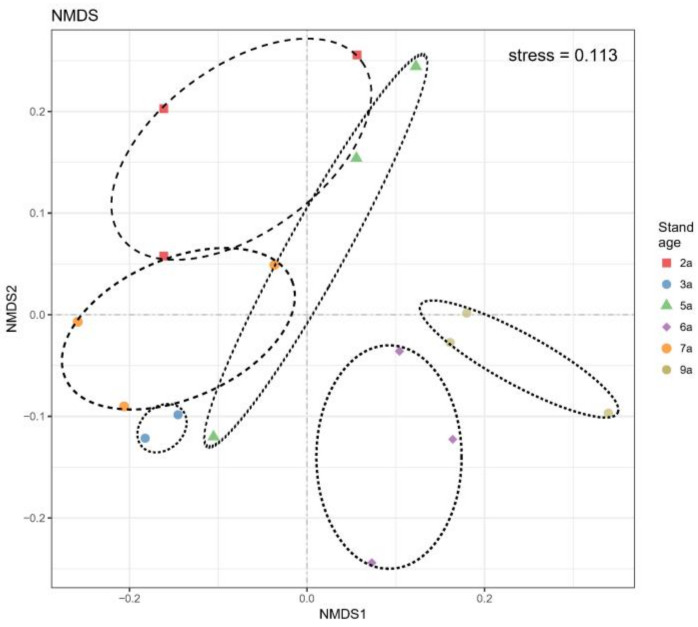
Non-metric multidimensional scaling plots of the variation in the AM fungal community, based on the profiles of the relative abundance of OTUs.

**Figure 6 jof-10-00404-f006:**
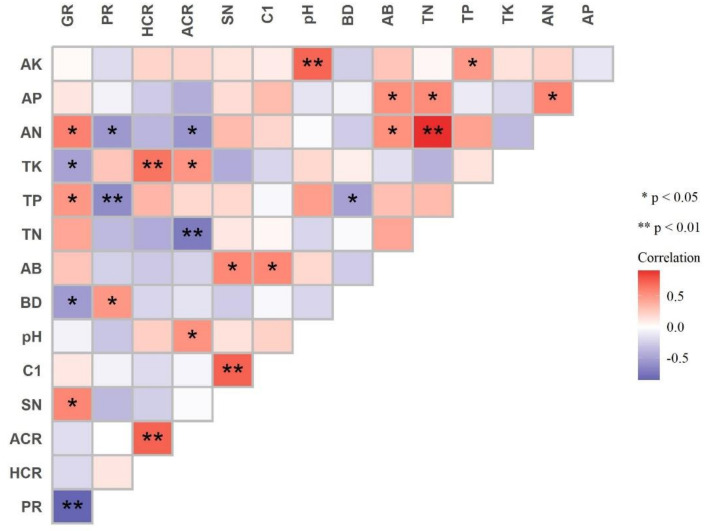
Pearson correlations between age-affected soil properties and AM fungal attributes in *Eucalyptus grandis* plantations. Note: *: significant at *p* < 0.05; **: significant at *p* < 0.01; PR: *Paraglomus* abundance; GR: *Glomus* abundance; HCR: hyphal colonization rate; ACR: arbuscular colonization rate; SN: Shannon index; C1: Chao1 index; pH: pH value; BD: bulk density; AB: available B; TN: total N; TP: total P; TK: total K; AN: available N; AP: available P; AK: available K.

**Figure 7 jof-10-00404-f007:**
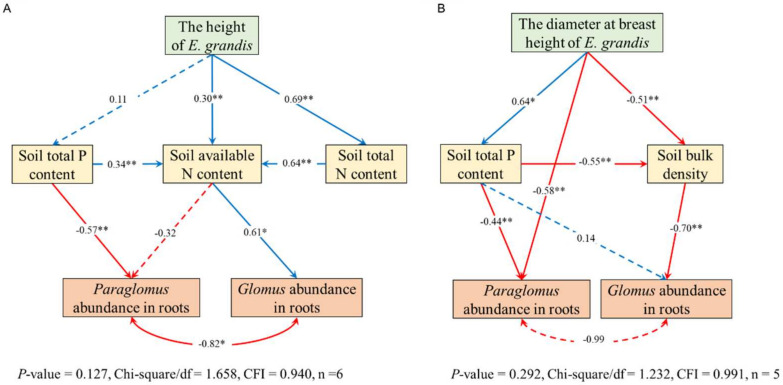
Structural equation model of relationships among age-affected growth statuses, soil properties, and AM fungal attributes in *Eucalyptus grandis* plantations. (**A**,**B**) show the model in 2–6 years plantations and 7–9 years plantations, respectively. Note: The lines among these parameters are the standardized regression weights (*: significant at *p* < 0.05; **: significant at *p* < 0.01). Red solid lines: the standardized regression weights are negative; blue solid lines: the standardized regression weights are positive; dotted lines: the standardized regression weights are insignificant (*p* > 0.05).

**Table 1 jof-10-00404-t001:** The soil properties of *Eucalyptus grandis* plantations with different stand ages.

Stand Age (Years)	pH Value	Soil Moisture (%)	Bulk Density	Sand Content % (2–0.05 mm)	Powder Content % (0.05– 0.002 mm)	Clay Content % (<0.002 mm)
2a	4.74 ± 0.22	0.20 ± 0.02	1.38 ± 0.03	46.33 ± 3.79	35.00 ± 5.20	18.67 ± 1.53
3a	4.22 ± 0.00	0.24 ± 0.04	1.36 ± 0.09	32.50 ± 4.95	42.00 ± 2.83	25.50 ± 2.12
5a	4.52 ± 0.03	0.22 ± 0.02	1.37 ± 0.08	31.33 ± 2.08	45.67 ± 4.16	23.00 ± 3.61
6a	4.65 ± 0.07	0.23 ± 0.03	1.30 ± 0.06	46.67 ± 17.24	31.00 ± 7.94	22.33 ± 9.50
7a	4.71 ± 0.33	0.22 ± 0.07	1.42 ± 0.11	44.67 ± 9.02	34.33 ± 3.21	21.00 ± 6.24
9a	4.55 ± 0.07	0.23 ± 0.03	1.32 ± 0.11	45.00 ± 5.57	33.67 ± 8.02	21.33 ± 2.89

Note: Data expressed as mean ± standard deviation.

## Data Availability

The datasets used to produce the main results are available upon reasonable request to the corresponding author.

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
