# Peer review of "Changes in the Arbuscular Mycorrhizal Fungal Community in the Roots of Eucalyptus grandis Plantations at Different Ages in Southern Jiangxi, China"

_jof, 2024, doi:10.3390/jof10060404_

Round 1
Reviewer 1 Report
The work represents a good contribution to the study of mycorrhizae in eucalyptus. However, its presentation needs to be improved.
Suggestions;
- The title should best represent the work
- The objective of the work is not well defined
- Keywords cannot be in the title
- Standardize the use of AM fungi or AMF throughout the text
- All scientific names used lack the author
- Material and methods lack information on diversity indices
-In the DNA analysis, in some steps the type of kit that was used was not mentioned
- The titles of the tables and graphs are incomplete flat scientific name fo eucalyptus, location
- It is not well put on why to value Boro
- The conclusions are more discussion than conclusion. Redraw conclusions based on objectives.
Reviewer 2 Report
The manuscript reports a study of plant growth, soil attributes, and arbuscular mycorrhizal fungal diversity in Eucalyptus plantations of different ages. The survey was all planned and was well executed. However, the work has some flaws. The authors must provide data on the soil attributes; stating they were similar is insufficient. The manuscript needs extensive revision, besides English language editing. The only data of interest in Figure 2 are Total P and Total K; all the others with no significant effects may be eliminated. The data is not necessary. The authors state that the diversity indexes used showed no difference, specifying the mean values for each of them.
The authors should deepen the Discussion on Figures 6 and 7, as the analyses are related and merit to have their relationship further explored.
As Eucalyptus plants also and predominantly form ectomycorrhizas, this aspect should be discussed in the manuscript.
The Conclusion wraps up the Discussion and should not be separated from it. The authors need to state their Conclusion concisely and directly.
Line 99: Replace “roots should be” with “were”.
Line 112: If the authors used a stain that uses phenol, the citation is correct; however, if they used a phenol-free stain, they should cite Koske and Gemma, 1989 (https://doi.org/10.1016/S0953-7562(89)80195-9),
Line 127: Citation needed for the primers used.
Lines 162-163: Revise the sentence for clarity.
Figure 2: Only the graphs regarding total P and total K should be in this Figure since all other variables showed no significant change with age.
Figure 3: The data about mycorrhizal colonization has some potential, but there is no physiological explanation for the curve shown for arbuscules. These structures are transient and quite sensitive to weather and soil conditions.
Table 1. This table is not necessary. The authors just state that the diversity indexes used showed no difference, specifying the mean values for each.
Reviewer 3 Report
The investigation of AMF composition in recovered eucalyptus forests of different ages is the focus of the reviewed work. The authors cite a number of recent articles that emphasize the critical function that AMF plays in the development of eucalyptus, particularly in stressed conditions like drought and heavy metal pollution.
When examining the article's contents, the first question that emerged was: What was the study's primary goal, and what is the novelty of the findings? Was the goal of the research to clarify which biotic (environmental) factors affect the number of eucalyptus plantations and how they interact with one another? Do these facts offer any potential benefits for growing eucalyptus? Although the history of the study site is not mentioned in the article, eucalyptus trees are typically chopped down every few years in planted woods. What is the primary research finding regarding the AMF analysis? Is it necessary to alter the cultivation scheme (tree growth period), fertilize (adding additional chemicals during fast or slow growth in a complex or alone), introduce an AMF inoculant at the start of the growth period, etc.?
The authors have taken into consideration the parameters of height and diameter as tree growth indicators (Section 3.1) and have incorporated them into the structural equation model. The reviewer believes that this is not warranted. As it stands, when compared to Pearson correlation analysis, the model's conclusions offer nothing novel. Sure, ages of eucalyptus stands change the physical and chemical characteristics of the soil and impact the ability of AMF to colonize, however it's unclear how these variations were detected in this model.
Several particular queries concerning the object and method descriptions arise.
1. How long have eucalyptus plantations been grown?
2. What kind of kit was used to extract DNA?
3. Would you kindly provide the primers that were used for amplification? Do they are universal?
4. Do than 50,000 (L.221) mean raw Illumina sequences or suitable AMF-sequences?
5. For each sample, how many libraries were examined? Are the figures 4's data averages?
Minor: 1. Make Glomus (LL.227,228, 248,etc.) italic.
2. Red Curve: What does mean (Fig.2)?
3. Fig. 3: The values along the ordinate axis are not labeled. In methods section hyphal colonization is described in %.
4. LL. 313-315 – unclear; could you kindly rewrite?
5. Paraglomerales (L.318)
6. LL. 325–326 - unclear
Reviewer 4 Report
The article, complete and well structured, reports innovative methods for the study of AM fungal population.
The following minor revisions are suggested: insert some photos of the mycorrhizal infection; take care of the italics of the names of the genera (e.g.: Glomus, line 227 and following).
Author Response
Dear Reviewer,
Thank you very much for your detailed review and constructive comments on our manuscript. We are pleased to hear your positive feedback regarding the structure and innovative methods of our study on AM fungal population.
Regarding your specific suggestions:
Insert photos of the mycorrhizal infection: While we appreciate the suggestion to include photos, we have chosen not to add them due to space constraints and to maintain the focus on the quantitative and methodological aspects of our study. Additionally, the microscopic images of mycorrhizal infection are well-documented in existing literature and are not central to the new methodologies we are presenting.
Take care of the italics of the names of the genera: We have thoroughly reviewed the manuscript and ensured that all genera names, such as Glomus, are correctly italicized throughout the text, including on line 227 and subsequent occurrences.
We appreciate your valuable feedback, which has significantly improved the quality of our paper. Should you have any further suggestions or require additional information, please do not hesitate to contact us.
Round 2
Reviewer 2 Report
The authors have made the changes that were recommended or suggested, improving the quality of the manuscript.
The authors eliminated unnecessary inforamtion form the Figures and Tables.
Reviewer 3 Report
The authors did a good job, responded in detail to the reviewers' comments and made the necessary corrections to the text.
No comments